# Time-to-event estimation of birth prevalence trends: A method to enable investigating the etiology of childhood disorders including autism

Alexander G. MacInnis ⬤ *

Independent Researcher, Los Altos, California, United States of America

* a.macinnis@alumni.stanford.edu

**Data Availability Statement:** Data for this paper are generated by simulation software, which is available at https://doi.org/10.17605/OSF.IO/WPNKU.

## Abstract

An unbiased, widely accepted estimate of the rate of occurrence of new cases of autism over time would facilitate progress in understanding the causes of autism. The same may also apply to other disorders. While incidence is a widely used measure of occurrence, birth prevalence—the proportion of each birth year cohort with the disorder—is the appropriate measure for disorders and diseases of early childhood. Studies of autism epidemiology commonly speculate that estimates showing strong increases in rate of autism cases result from an increase in diagnosis rates rather than a true increase in cases. Unfortunately, current methods are not sufficient to provide a definitive resolution to this controversy. Prominent experts have written that it is virtually impossible to solve. This paper presents a novel method, time-to-event birth prevalence estimation (TTEPE), to provide accurate estimates of birth prevalence properly adjusted for changing diagnostic factors. It addresses the shortcomings of prior methods. TTEPE is based on well-known time-to-event (survival) analysis techniques. A discrete survival process models the rates of incident diagnoses by birth year and age. Diagnostic factors drive the probability of diagnosis as a function of the year of diagnosis. TTEPE models changes in diagnostic criteria, which can modify the effective birth prevalence when new criteria take effect. TTEPE incorporates the development of diagnosable symptoms with age. General-purpose optimization software estimates all parameters, forming a non-linear regression. The paper specifies all assumptions underlying the analysis and explores potential deviations from assumptions and optional additional analyses. A simulation study shows that TTEPE produces accurate parameter estimates, including trends in both birth prevalence and the probability of diagnosis in the presence of sampling effects from finite populations. TTEPE provides high power to resolve small differences in parameter values by utilizing all available data points.

## Introduction

An unbiased, widely accepted estimate of the rate of occurrence of new cases of autism over time would facilitate progress in understanding what causes autism. It would also help answer

**Funding:** The author(s) received no specific funding for this work.

**Competing interests:** The authors have declared that no competing interests exist.

other important questions, such as projecting future adult caseloads based on cases already born. Epidemiology relies on the rate of occurrence of new cases (occurrence rate) to investigate the risks and causes of any disorder or disease [1, 2]. However, studies of autism epidemiology commonly speculate that estimates showing strong increases in the occurrence of autism are the result of an increase in rates of diagnosis rather than a true increase in cases [3–19]. Current methods are not sufficient to provide a definitive resolution to this controversy, as the Background and Literature Review section explains. Prominent experts have written that it is virtually impossible to solve [20].

This paper presents a novel analytical method called time-to-event birth prevalence estimation (TTEPE) to estimate birth prevalence trends. It quantifies the occurrence rate while avoiding the biases and ambiguity of existing methods. This paper focuses on autism, but the method may also benefit various diseases, disorders, and conditions.

There are multiple specific measures of the occurrence rate. **Birth prevalence** is the preferred measure for birth defects, congenital diseases, and disorders—including autism—where the times of occurrence events are not observable because occurrence is before or shortly after birth [2, 21–25]. Birth prevalence is also known as birth year prevalence and birth year cohort prevalence. Birth prevalence is the proportion of a birth year cohort that has the disorder. A birth year cohort is the set of all individuals born in a given year. While conventional estimates of birth prevalence are based on counting diagnoses, the definition of birth prevalence does not depend on when each case was diagnosed or whether each case was ever diagnosed. **Incidence** is based on the times of occurrence of new cases and is a measure commonly used to describe the occurrence rate [1, 2]. The incidence of diagnoses (i.e., the rate of occurrence of new diagnoses) is different from the incidence of cases. The difference between incident (new) diagnoses and incident (new) cases is fundamental when considering the hypothesis that increasing rates of diagnosis reflect improved ascertainment of cases rather than an increasing case occurrence rate.

There are currently very few studies that estimate the birth prevalence of autism, and there is no generally accepted estimate of this measure [3–16]. Numerous papers instead examine autism **prevalence**, which is a measure of the total number of cases at a specific time, not the occurrence rate. Prevalence is rarely useful in studying etiology [2, 13, 14]. Estimates of both prevalence and birth prevalence of autism remain controversial, with no known method of resolution. The next section provides essential details.

While the term "birth prevalence" contains the word "prevalence," its meaning differs from the standalone term. "Prevalence" on its own is not specific to birth year and is specific to time. Comparison of prevalence estimates across multiple studies is generally not suitable for understanding trends in incidence or birth prevalence [2]. Different prevalence estimates may use different case definitions, case-finding procedures, and ranges of ages, among other possible differences between prevalence studies [13]. **Cumulative incidence** is the sum of events, such as incident diagnoses, as proportions of the cohort, from birth to a specified age. Cumulative incidence of diagnoses approximates birth prevalence, but the result can be biased [1, 26–28, S1 Text].

**Diagnostic factors** are factors that affect the probability of diagnosing cases. **Probability of diagnosis** is the probability of diagnosing individual cases with the disorder who are exhibiting diagnosable symptoms. This probability may vary with time. When the probability of diagnosis increases, a greater proportion of cases are diagnosed; that is, there is increased ascertainment of cases. Diagnostic factors include, among others: awareness, outreach efforts, screening, diagnostic practice, diagnostic substitution or accretion, availability of evaluations, diagnostic criteria, social factors, policies, and financial incentives for diagnosis. The Method section below shows that the probability of diagnosis is a function of the set of all diagnostic

factors. **Diagnostic criteria** specify the set of symptoms that qualify an individual for a diagnosis [29, 30] and can affect the effective birth prevalence of each birth cohort. Formal diagnostic criteria change at specific dates when new criteria take effect.

The TTEPE method is specifically designed to produce valid, accurate estimates of birth prevalence properly separated from the effects of diagnostic factors and diagnostic criteria. A suitable method to establish birth prevalence should:

- estimate birth prevalence over a broad range of birth years;

- adjust for the effect of the set of all diagnostic factors;

- adjust for changes in diagnostic criteria; and

- disentangle birth prevalence, diagnostic factors, and diagnostic criteria such that the adjustments do not introduce significant bias or uncertainty in the estimates.

The following sections give a concise review of relevant studies followed by the presentation of TTEPE. TTEPE extends established time-to-event or survival analysis to model the effects of diagnostic factors and diagnostic criteria on rates of incident diagnoses by age and year of diagnosis. It estimates birth prevalence over multiple cohorts. The method presentation depicts the assumed causal paths, derives the model's formulas, lists the assumptions underlying the analysis, discusses potential violations of the assumptions, and presents validation of the method via simulation, followed by a discussion.

## Background and literature review

Broad-based literature reviews by Elsabbagh [14] and Fombonne [13] confirmed rises in reported autism prevalence over time. However, both stated that what matters is the number of new cases over time (i.e., the occurrence rate) and that rising prevalence does not necessarily imply an increase in the rate of new cases, consistent with textbooks [1, 2].

A few papers provide estimates of autism birth prevalence over time, in some cases using different names. A series of reports from the US Centers for Disease Control and Prevention's (CDC) Autism and Developmental Disabilities Monitoring Network (ADDM) [3–11] estimate what they call the prevalence of autism among children who were eight years old at each even-numbered year 2000 through 2016. These are actually birth prevalence estimates, but the reports describe the findings as simply "prevalence" with no mention of birth prevalence. The ADDM reports show a strong trend in birth prevalence by birth year and speculate that the observed increase may result from various factors other than a true increase [3–11]. Meyers [12] used a records-based review, largely similar to the ADDM approach, as well as counting community diagnoses. The authors found a sharp increase in cumulative incidence by birth year, approximating birth prevalence, and suggested that the increase may result from factors such as increasing awareness or broadening diagnostic criteria rather than a true increase [12]. Hansen [26] is a methods paper that recommends using the cumulative incidence of diagnoses of childhood psychiatric disorders for each birth cohort as a measure of risk when risk may vary by birth year. They found a strongly increasing cumulative incidence of autism by birth year cohort but made scant mention of it. Sasayama [16] found an increasing cumulative incidence of autism to age five by birth year and suggested it may have been caused by increasing awareness. None of these papers suggest methods to quantify or adjust for the effects of diagnostic factors or diagnostic criteria [3–16, 20, 21, 26].

Supplementary file [S1 Text] shows that while one can use cumulative incidence to estimate the birth prevalence trend, the result is potentially ambiguous. It is possible to explain a trend

in cumulative incidence by birth year due to any of a wide range of combinations of the trends in birth prevalence and diagnostic factors; see also MacInnis [27].

Croen [31] examined birth prevalence trends in autism and mental retardation in California over birth years 1987 to 1994. The authors concluded that the data and methods available were insufficient to determine how much of the observed increase reflected a true increase in birth prevalence. Nevison [32] presented California Department of Developmental Services data showing a sharp rise in the birth prevalence of autism over several decades. They discuss various non-causal hypotheses that others have suggested to explain the observed trend but did not discuss methods to quantify or adjust for the hypothesized factors.

Various papers have tried to adjust or control for diagnostic factors or diagnostic criteria to estimate case trends using one of two closely related methods. One approach adjusts for variables representing the effects of diagnostic factors, diagnostic criteria or both when estimating time trends in case rates. The other is age-period-cohort analysis, which typically uses regression with a linear predictor containing terms for age (at diagnosis), period (year of diagnosis), and cohort (birth year). Neither method is suitable; both methods either introduce bias or produce unreliable results, as explained here. Campbell [15] and McKenzie [33] both point out that various factors could potentially affect the rate of diagnoses without affecting the true case rate. Elsabbagh, Campbell, and Baxter [14, 15, 34] suggest controlling for changes in diagnostic factors or diagnostic criteria to enable the comparison of prevalence estimates to estimate time trends. Elsabbagh [14] wrote: "Time trends in rates can therefore only be gauged in investigations that hold these parameters under strict control over time," referring to case definition (diagnostic criteria) and case ascertainment (probability of diagnosis). However, none of the papers we have found proposes a method that avoids introducing bias while controlling for the indicated parameters. All three variables—the probability of diagnosis, diagnostic criteria, and birth prevalence—are functions of time, so none of them remain constant when any of them changes, creating a fundamental problem with attempting to control for any of these variables. Baxter [34] adjusted for dichotomous variables representing the most recent diagnostic criteria, each of which took effect at a specific date, assuming that these variables introduced bias. However, such adjustment is problematic, and itself introduces bias. Schisterman [35] shows that controlling for variables on a causal path from the input (time, in this case) to the outcome (prevalence or incident diagnoses) constitutes inappropriate adjustment and biases the estimate of the primary effect (i.e., of time on the outcome) towards zero. Similarly, Rothman [2] states that controlling for intermediate variables typically causes a bias towards finding no effect.

Other papers used age-period-cohort analysis to attempt to separate the effects of diagnostic factors from either birth prevalence trends or changes in prevalence. Examples include Keyes [17], King [18], and MacInnis [27]. However, this approach cannot produce valid estimates except in very special circumstances [27, 36, 37, S1 Text]. The root of the problem is the exact collinearity (i.e., cohort + age = period) which violates a basic assumption of regression and causes the model to be unidentified. That is, there is an infinite number of parameter set values such that estimation could produce any arbitrary one of them [38]. One can constrain the analysis to make the model identified; however, the constraint imposes an a priori assumption on the solution [27, 36, 37, S1 Text]. Keyes [17] used the constraint approach, and Spiers [39] pointed out that the method used could as easily have reached the opposite conclusion. King [18] assumed that period effects are dominant and controlled for birth year, forming a constrained age-period-cohort analysis. Analysis that implicitly assumes little to no trend in birth prevalence causes the estimate to fit the assumption [27, 37]. Analyses that omit one or two of the age, period, and cohort variables implicitly assume that the coefficients of the omitted variables are equal to zero, which might not be valid. That approach does not solve the problems

of age-period-cohort analysis [36, 37, S1 Text]. It is not suitable to estimate the age trend directly from the data because such estimates make implicit assumptions about either the period or cohort trends and result in conclusions that follow the assumptions [27].

We did not find any papers that use survival or time-to-event analysis in ways that are directly relevant to this paper. The Discussion section gives details.

This paper follows advice from Schisterman [35], who recommends "clearly stating a causal question to be addressed, depicting the possible data generating mechanisms using causal diagrams, and measuring indicated confounders."

## Method of time-to-event birth prevalence estimation (TTEPE)

### Description of the method

TTEPE estimates birth prevalence over a range of birth years, properly adjusted for the effects of diagnostic factors and changes in diagnostic criteria, in a way that solves the problems of previous methods. TTEPE input data provides either the ages or years of initial diagnoses; this information exists in some datasets [12, 16, 19, 28, 40–43]. One key element is recognizing that the probability of diagnosis at the year of each initial diagnosis (diagnostic year) represents the effect expressed in the data of the set of all diagnostic factors, as shown in the next section and introduced in [27]. The problem is then one of separating the effects of birth year and diagnostic year, called cohort and period, respectively, in age-period-cohort analysis. Age-period-cohort analysis inherently involves age (at diagnosis), and we must properly account for the effect of age without using a linear predictor in age, period, and cohort. Therefore, another fundamental element of TTEPE is modeling the distribution of initial diagnoses with age—the age distribution—as a separate, non-linear process. The age distribution may differ between birth cohorts. The age distribution may ramp up from age zero to a peak at some age and fall approximately exponentially with further increasing age. TTEPE models the rising distribution from age zero as cases developing diagnosable symptoms with age, called eligibility. It models the falling distribution of initial diagnoses with age using a survival process defined below. TTEPE incorporates variables that model the effect of changes in diagnostic criteria. Optimization software finds the parameter values that result in the model best fitting the observed data, thereby estimating birth prevalence and the other variables of interest. This paper explains all of these in detail.

In TTEPE's survival process, "survival" refers to cases that have not yet been diagnosed. In each cohort, there is some unknown proportion that are cases—the birth prevalence. As the cases develop diagnosable symptoms—become eligible—they join the pool of cases at risk of initial diagnosis, called the risk set. The size of the risk set as a proportion of the cohort is called $R$. At each age, some portion of the cohort's cases become eligible, thereby increasing $R$, and some portion of the risk set is diagnosed, reducing $R$. At each age, $R$ multiplied by the probability of diagnosis gives the proportion of the cohort receiving an initial diagnosis. The reduction of $R$ due to these diagnoses results in fewer cases being diagnosed at the next age. Thus, this process inherently produces a falling number of diagnoses with increasing age once most or all cases are old enough to be eligible.

The birth prevalence $BP$ of each cohort determines the magnitude, not the shape, of the age distribution. The shape of the age distribution at each cohort is determined by the probability of diagnosis $PD$ and eligibility $E_A$, which is a function of age. Greater values of $PD$ cause a greater proportion of the risk set to be diagnosed at each age, leaving a smaller proportion of cases at risk at the next age. Eligibility $E_A$ is the proportion of cases exhibiting diagnosable symptoms at age $A$. Few, or no cases may be eligible at birth, and cases may develop diagnosable symptoms with increasing age, thereby increasing $E_A$ up to a maximum value of 1. If the

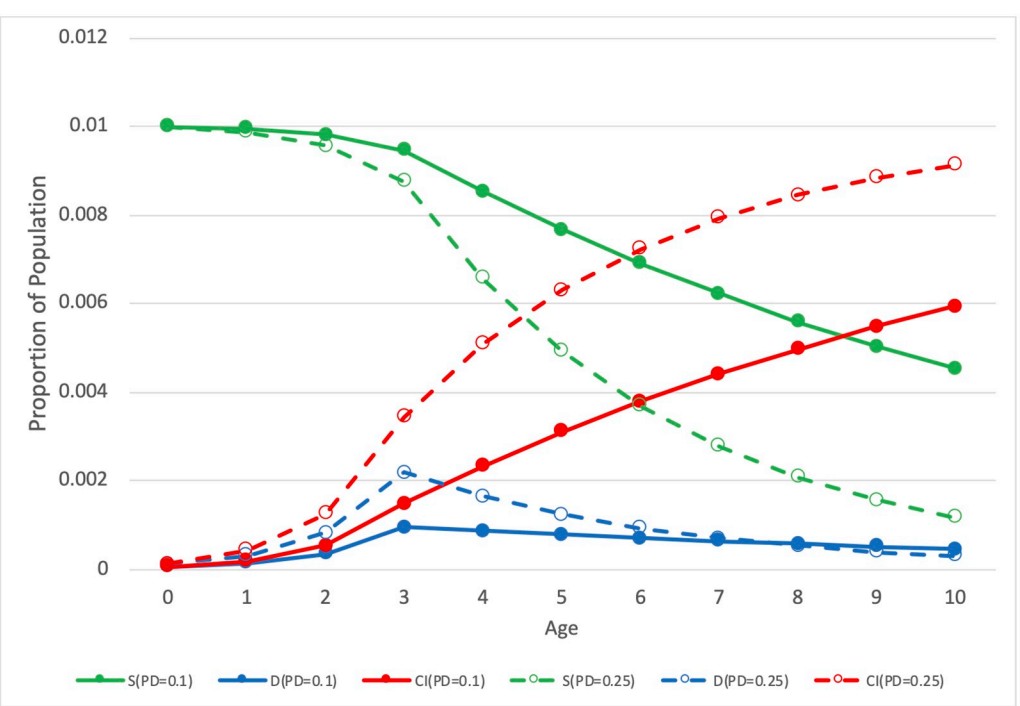

**Fig 1. Example of a survival process for two values of probability of diagnosis *PD*.** The green lines *S* denote survival, the blue lines *D* denote the rate of diagnoses, and the red lines *CI* denote cumulative incidence. The solid lines represent *PD* = 0.1, and the dashed lines represent *PD* = 0.25.

disorder's definition requires that all cases exhibit diagnosable symptoms by a certain age, then $E_A$ becomes equal to 1 at that age.

Fig 1 illustrates an example of the survival process that models the age distribution. It shows two scenarios with different values of the probability of diagnosis: *PD* = 0.1 and *PD* = 0.25. In this example, birth prevalence *BP* is 0.01 (1%) in both scenarios and eligibility $E_A$ increases from 0 at age 0 to 1 at age 3. The model produces predicted values of the proportion *D* of the cohort receiving initial diagnoses at each age, corresponding to the input data. The figure shows the survival function *S*, which is the proportion of the cohort consisting of undiagnosed cases at each age, regardless of their eligibility, unlike *R*, which incorporates eligibility. $S > R$ at ages where some cases are not yet eligible ($E_A < 1$) and $S = R$ at ages where all cases are eligible ($E_A = 1$). Cases not yet eligible are surviving (not yet diagnosed), but they are not yet in the risk set. For clarity, the figure shows *S* rather than $E_A$ and *R*. The shape of *R* (not shown) tracks the shape of *D*, which is determined by *PD* and $E_A$, however, *R* is at a different scale. Cumulative incidence *CI* shows how many cases have been diagnosed by each age. Note that greater values of *PD* result in both a left shift of the curve of initial diagnoses towards younger ages and a greater cumulative incidence at the end of follow-up, which in this example is age 10. While generally, *PD* varies as a function of diagnostic year, this example uses constant values of *PD* for clarity.

Later sections explain details of the method, including eligibility and the effect of changes in diagnostic criteria.

TTEPE is particularly applicable to disorders where case status is either present or predetermined by birth or a known age. Diagnosable symptoms may be present by some consistent age, or cases may develop diagnosable symptoms gradually over a range of ages, depending on the disorder.

## Method details

Input data comprises rates of initial diagnoses $D$ by age or diagnostic year for each birth year cohort. Since birth year + age = diagnostic year, these variables are interchangeable, subject to truncation or rounding. The rate is the count of initial diagnoses divided by the cohort's population at the respective diagnostic year or age.

TTEPE models the rate of initial diagnoses as a function of birth year, diagnostic year, and age. $D(BY, DY, A) = fmodel(BP_{BY}, PD_{DY}, CF_{DY}, E_A)$ where $D$ is the proportion of the birth cohort that receives a diagnosis, $BY$ is birth year, $DY$ is diagnostic year, $A$ is age, $BP$ is birth prevalence, $PD$ is the probability of diagnosis, $CF$ is the criteria factor, and $E$ is eligibility. The non-linear function $fModel$ is specified below. Importantly, $BP_{BY}$ comprises all cases in cohort $BY$ regardless of when or whether they have been diagnosed. The probability of diagnosis $PD_{DY}$ is the probability of diagnosing eligible, undiagnosed cases. $PD_{DY}$ is the effect of the set of all diagnostic factors on the data, as explained below. The criteria factor $CF_{DY}$ modifies the proportion of each cohort that qualifies as cases according to each new criteria set that takes effect at diagnostic year $DY$. $CF = 1$ for the criteria in effect at the first $DY$. Eligibility $E_A$ is the proportion of cases that have developed diagnosable symptoms at age $A$. The following text and causal directed acyclic graph (DAG) in Fig 2 explain and illustrate the paths from $BY$, $DY$ and $A$ to new diagnoses, the model outcome. The model can either specify or estimate the eligibility function $E_A$.

Diagnostic factors are those that affect the diagnosis of eligible cases. Any factor that affects the probability of diagnosis is a diagnostic factor; the introduction lists examples. $PD$ is equivalent to the hazard $h$ in discrete time-to-event or survival analysis. For each case of the disorder, the information resulting from $PD_{DY}$ consists of the time (diagnostic year) of initial diagnosis. $PD_{DY}$ has no effect on the data before diagnosing each case, none after the initial diagnosis since TTEPE considers only initial diagnoses, and none if the case is not diagnosed by the end of follow-up. Hence, $PD_{DY}$ is the effect of the set of diagnostic factors on the model outcome and is a function of diagnostic year.

The directed acyclic graph (DAG) in Fig 2 illustrates the causal paths from birth year $BY$, diagnostic year $DY$, and age $A$ to initial diagnosis $D$. Birth year is the independent variable in birth prevalence $BP$. Birth year $BY$ drives variable etiologic (causal) factors, which produce the disorder. Eligibility $E$ represents the development of diagnosable symptoms with age. Diagnostic criteria determine whether each individual's symptoms qualify them as a case. Changes in diagnostic criteria at specific years can change the proportion of the cohort classified as cases,

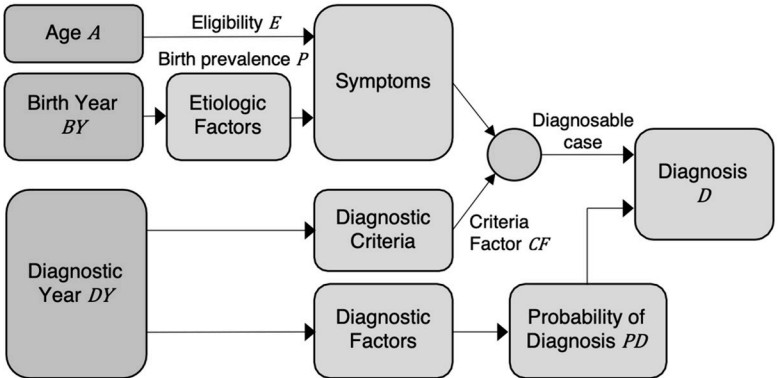

**Fig 2. Directed Acyclic Graph (DAG) showing the effects of birth year, diagnostic year and age on rates of diagnoses.**

i.e., the effective birth prevalence, by modifying the threshold of symptoms that qualify case status. This effect is represented by the criteria factor $CF$, which is a function of diagnostic year $DY$. Symptoms combined with diagnostic criteria form diagnosable case status. $DY$ is the independent variable in the effects of diagnostic factors, which manifest as the probability of diagnosis $PD$. Diagnosable case status and probability of diagnosis operate jointly to produce each initial diagnosis.

## TTEPE time-to-event analysis model

TTEPE is based on the DAG of Fig 2 and the survival process described above. For each birth cohort, undiagnosed eligible cases form the risk set, the size of which is denoted $R$. $BP$ is the proportion of the cohort that consists of cases, which is unknown and is the primary target of estimation, in contrast to known prior uses of survival analysis. $R$ is a function of $BP$, eligibility $E$, and the survival process. A later section describes using the effective birth prevalence $EBP$ instead of $BP$ to incorporate the criteria factor $CF$.

A portion of the risk set is diagnosed at each age $A$ according to the probability of diagnosis $PD$. The value of $PD$ at each age, or equivalently each diagnostic year, for each cohort, is the number of newly diagnosed cases divided by $R$. This formulation applies because TTEPE uses a discrete-time model; see Kalbfleisch [44]. At each age, newly eligible cases are added to the risk set, and newly diagnosed cases are removed from the risk set. For any given value of $PD$, as $R$ decreases or increases, the population proportion of initial diagnoses $D$ changes accordingly. This process generates the age distribution of diagnoses.

Let $D_{BY,A}$ be the population-based rate of incident (first) diagnoses. The model generates predicted values $\widehat{D_{BY,A}}$. Modeling proportions rather than counts accommodates changes in each cohort's population size over time, e.g., due to in- and out-migration and deaths. Alternatively, the analysis could model counts directly.

Let $PD_{DY}$ be the probability of diagnosis at diagnostic year $DY$. $PD_{DY}$ is equivalent to $PD_{BY,A}$ because $DY = BY + A$, subject to truncation or rounding. Let $BP_{BY}$ be the birth prevalence of cohort $BY$, i.e., the proportion of the cohort consisting of cases regardless of how many have been diagnosed. $BP_{BY}$ does not depend on eligibility. Let $R_{BY,A}$ be the discrete risk set, which is the proportion of birth year cohort $BY$ consisting of eligible cases at risk of initial diagnosis at age $A$. Let $E_A$ be the discrete eligibility function, the proportion of cases that are eligible at age $A$, bounded by $0 \leq E \leq 1$. At each age $A \geq 1$, $BP \times (E_A - E_{A-1})$ is the incremental portion of $BP$ added to $R$ due to increases in eligibility. For simplicity, assume $E_A$ increases monotonically (non-decreasing), meaning that cases do not lose eligibility before diagnosis. At each age $A$ for each cohort $BY$, the rate of incident diagnoses $D_{BY,A} = R_{BY,A} \times PD_{DY}$ from the definition of $PD$ above. We write $PD_{DY}$ as $PD_{BY,A}$ to specify the model in terms of $BY$, noting that $DY = BY + A$, subject to truncation or rounding. Kalbfleisch [44] gives background on general time-to-event theory and equations.

**Derivation of formulas.** First, consider the simplest scenario, where all cases are eligible for diagnosis from birth (constant $E_A = 1$ for all values of $A$), and the diagnostic criteria do not change the effective birth prevalence over the interval of interest. In the second scenario, $E_A$ increases monotonically, and in the third scenario, $E_A$ increases until it plateaus at $E_A = 1$ for $A \geq AE$, where $AE$ is the **age of complete eligibility.** A later section adds the effect of changes in diagnostic criteria.

<u>Constant $E_A = 1$.</u> For $A \geq 1$, $E_A - E_{A-1} = 0$. For the first year of age, $A = 0$, the risk set $R_{BY,0} = BP_{BY}E_0 = BP_{BY}$ and $D_{BY,0} = R_{BY,0}PD_{BY,0} = BP_{BY}PD_{BY,0}$. In other words, at age 0, all cases are eligible and in the risk set, and the proportion of the cohort (not the proportion of cases) that is diagnosed is the birth prevalence $BP$ times the probability of diagnosis $PD$ at age 0.

For $A = 1$, $R_{BY,1} = BP_{BY} - D_{BY,0} = BP_{BY} - BP_{BY}PD_{BY,0} = BP_{BY}(1 - PD_{BY,0})$ and $D_{BY,1} = R_{BY,1}PD_{BY,1} = BP_{BY}(1 - PD_{BY,0})PD_{BY,1}$.

That is, the size of the risk set decreases from age 0 to age 1 by the proportion of the cohort diagnosed at age 0. The proportion of the cohort diagnosed at age 1 is the size of the risk set at age 1 times the probability of diagnosis at age 1.

For $A = 2$, $R_{BY,2} = R_{BY,1} D_{BY,1} = BP_{BY}(1 - PD_{BY,0}) - BP_{BY}(1 - PD_{BY,0})PD_{BY,1} = BP_{BY}(1 - PD_{BY,0})(1 - PD_{BY,1})$ and $D_{BY,2} = R_{BY,2} = R_{BY,2}PD_{BY,2} = BP_{BY}(1 - PD_{BY,0})(1 - PD_{BY,1})PD_{BY,2}$.

Similarly for $A = 3$, $R_{BY,3} BP_{BY}(1 - PD_{BY,0})(1 - PD_{BY,1})(1 - PD_{BY,2})$ and $D_{BY,3} = BP_{BY}(1 - PD_{BY,0})(1 - PD_{BY,1})(1 - PD_{BY,2})PD_{BY,3}$.

Combine these expressions and generalize to, for $A \geq 1$,

$$R_{BY,A} = BP_{BY} \prod_{a=0}^{A-1} (1 - PD_{BY,a})$$

and

$$D_{BY,A} = BP_{BY} \prod_{a=0}^{A-1} (1 - PD_{BY,a}) PD_{BY,A} \qquad (1)$$

In all three scenarios in this section, the survival function is:

$$S_{BY,A} = BP_{BY} - \sum_{a=0}^{A-1} D_{BY,a} \qquad (2)$$

$S_{BY,A}$ is the portion of $BP_{BY}$ that has not been diagnosed by age $A$. As noted above, $S$ differs from $R$ because of eligibility. The summation term is the cumulative incidence of initial diagnoses through age $A - 1$.

<u>Increasing $E_A$</u>. In this scenario, $E_0 < 1$ and $E_A$ increases monotonically with $A$. For $A = 0$, $R_{BY,0} = E_0 BP_{BY}$ and $D_{BY,0} = E_0 BP_{BY} PD_{BY,0}$. For each $A \geq 1$, $R_{BY,A} = R_{BY,A-1} - D_{BY,A-1} + (E_A - E_{A-1})BP_{BY}$. The last term represents the increase in $R_{BY,A}$ due to the incremental increase of $E_A$. Then,

$$D_{BY,A} = R_{BY,A} PD_{BY,A} = (R_{BY,A-1} - D_{BY,A-1})PD_{BY,A} + (E_A - E_{A-1})BP_{BY}PD_{BY,A} \qquad (3)$$

Eq (3) serves as a procedural definition for software implementing *fModel* where $E_A$ may increase from A = 0 to the maximum age of follow-up A = M. $E_M$ should be set to 1 to avoid creating an unidentified model. Equivalent expressions for $R_{BY,A}$ and $D_{BY,A}$ similar to Eq (1) follow, where each expression in the summations describes the portion of $BP_{BY}$ that becomes eligible at each age according to incremental increases in $E_A$. For $A \geq 1$,

$$R_{BY,A} = \sum_{a=0}^{A-1} (E_a - E_{a-1})BP_{BY} \prod_{b=a}^{A-1} (1 - PD_{BY,b})$$

$$D_{BY,A} = \sum_{a=0}^{A-1} (E_a - E_{a-1})BP_{BY} \prod_{b=a}^{A-1} (1 - PD_{BY,b}) PD_{BY,A} \qquad (4)$$

where $E_{-1}$ is defined to be 0. $E_A$ can be defined parametrically or non-parametrically.

<u>Plateau $E_A$</u>. In this scenario, $E_A$ increases with age from $E_0 < 1$ and plateaus at $E_A = 1$ for $A \geq AE$, $AE < M$, and $M$ is the maximum age of follow-up. External information, such as the disorder's definition, may indicate the value of $AE$, or investigators may specify $AE$ based on estimates of $E_A$ found using Eqs (3) or (4). For $A \leq AE$, Eq (3) applies as does the formula for

$R_{BY,A}$ preceding it. For $A > AE$, $(E_A - E_{A-1}) = 0$, and

$$D_{BY,A} = R_{BY,A}PD_{BY,A} = (R_{BY,A-1} - D_{BY,A-1})PD_{BY,A} \tag{5}$$

Alternatively, combine Eq (2) with the fact that $E_{AE} = 1$ to obtain
$R_{AE} = S_{AE} = BP_{BY} - \sum_{a=0}^{AE-1} D_{BY,a}$. Then,

$$D_{BY,AE} = R_{BY,AE}PD_{BY,AE} = S_{BY,AE}PD_{BY,AE} = \left(BP_{BY} - \sum_{a=0}^{AE-1} D_{BY,a}\right)PD_{BY,AE} \tag{6}$$

And for $A > AE$,

$$R_{BY,A} = S_{BY,A} = R_{BY,AE} \prod_{a=AE}^{A-1} (1 - PD_{BY,a})$$

$$D_{BY,A} = R_{BY,AE} \prod_{a=AE}^{A-1} (1 - PD_{BY,a})PD_{BY,A} \tag{7}$$

The form in Eqs (6) and (7) uses empirical values of $D_{BY,A}$ for $A < AE$ rather than modeling $D_{BY,A}$ from $BP_{BY}$, $E_A$ and $PD_{BY,A}$ using Eq (3). Eqs (6) and (5) serve as procedural definitions in models that assume $AE$ and do not estimate $E_A$, while Eqs (3) and (5) serve as procedural definitions in models that do estimate $E_A$.

The scenario of increasing $E_A$ is a general formulation. The scenario of plateauing $E_A$ may be appropriate when investigators have reason to specify $AE$. For example, for disorders where diagnosable symptoms are present by age three by definition, the plateau $E_A$ scenario applies, and the age of complete eligibility $AE = 3$.

**Birth prevalence, cumulative incidence and censoring.** The birth prevalence $BP$ of each cohort $BY$ is equivalent to the sum of the cumulative incidence of diagnoses through the last age of follow-up plus the portion of cases that were not diagnosed by the end of follow-up, i.e., the right-censored portion.

In all three scenarios above of $E_A$, we can express $BP$ as a function of the survival function $S_A$ and the cumulative incidence $CI_{A-1}$ for $A > 0$, by rearranging Eq (2) as $BP = S_A + CI_{A-1}$. Assuming eligibility $E_M = 1$ at the last age of follow-up $M$, then $S_M = R_M$. Then, $BP = R_M + CI_{M-1}$ and $D_M = R_MPD_M$. The censored proportion is $S_{M+1} = S_M - D_M$, which is equivalent to $S_{M+1} = R_M - R_MPD_M = R_M(1 - PD_M)$. After estimating the model parameters, the estimated censored proportion—the proportion of the cohort consisting of cases that have not been diagnosed by the last age of follow-up $M$—is $\widehat{S_{M+1}} = \widehat{R_M}(1 - \widehat{PD_M})$. $\widehat{R_M}$ is an internal variable in the model. Diagnoses are counted from birth, so there is no left censoring.

## Assumptions

The TTEPE method relies on several baseline assumptions. If some assumptions are not met, there could be bias in estimation results. Investigators can accommodate deviations from assumptions in many cases, as discussed in a later section. The TTEPE method does not assume any particular relationship between parameter values, nor does it require assuming the values of any explicit or implicit variables. The TTEPE method's baseline assumptions are:

1. The eligibility function $E_A$ is consistent across cohorts.

2. The probability of diagnosis *PD* applies equally to all eligible undiagnosed cases at any given diagnostic year.

3. The birth prevalence within each cohort under consistent diagnostic criteria is constant over the range of ages included in the analysis.

4. Data represent truly initial diagnoses.

5. Case status is binary according to the applicable diagnostic criteria.

6. The discrete-time interval (e.g., one year) is small enough that the error introduced by treating the variable values as constant within each interval is negligible.

7. There are no false positives in the dataset.

8. Any difference in competing risks between cases and non-cases in the range of ages analyzed is small enough to be ignored. Competing risk is a term of art in survival analysis. An example competing risk is death before initial diagnosis.

The assumption of a consistent eligibility function means that cases develop diagnosable symptoms as a function of age, and that function is the same for all cohorts. In other words, the value of $E_A$ at age $A$ is the same for all cohorts $BY$, while $E_A$ varies with $A$. The section Changes in criteria affecting birth prevalence discusses a separate effect that might make the eligibility function appear to be inconsistent even if it is not. A later section explores potential violations of assumptions.

## Estimating parameters

TTEPE performs a non-linear regression that estimates the parameters of the model of $D_{BY,A}$ using general-purpose non-linear optimization software. The model is based on one or more of Eqs (3), (5) and (6) selected based on the eligibility scenario. The model produces estimated values of $\widehat{D_{BY,A}}$ from the parameters and independent variables, and the software finds the parameter values that minimize a cost function $\text{cost}(D, \widehat{D})$. One suitable implementation of optimization software in the Python language is the curve_fit() function in the SciPy package (scipy.optimize.curve_fit in SciPy v1.7). Its default cost function is $(D - \widehat{D})^2$, so it minimizes the sum of squared errors. Python software to perform this regression and the simulations described below is available at OSF [45].

Investigators should choose which model equation to use based on knowledge or estimates of the eligibility function $E_A$. Non-parametric estimates $\widehat{E_A}$ can inform a choice of a parametric form of $E_A$. If $E_A = 1$ for all $A \geq AE$, that fact and the value of $AE$ should be apparent from estimates $\widehat{E_A}$, and the plateau $E_A$ scenario applies.

Investigators should choose forms of $BP_{BY}$ and $PD_{BY,A}$ appropriate to the dataset. Linear, first-order exponential, second-order exponential or non-parametric models may be appropriate. Graphical and numeric model fit combined with degrees of freedom can guide the optimum choice of a well-fitting parsimonious model.

If the population proportion of cases represented in the data is unknown for all cohorts, then absolute birth prevalence, or the intercept, may be underestimated by an unknown scale factor. Proportional changes in birth prevalence between cohorts are unaffected by underestimation of the intercept. Changes over time in the proportion of cases included in the sample reflect changing diagnostic factors, and the parameter estimates of *PD* automatically represent such changes. If the proportion of cases represented in the data is known for at least one cohort, one can use that knowledge to determine the intercept.

## Changes in diagnostic criteria affecting birth prevalence

Changes in criteria may change the effective birth prevalence *EBP* within a cohort, without affecting symptoms or etiology, by including or excluding as cases some portion of the cohort population compared to prior criteria. This mechanism is distinct from a changing probability of diagnosis. A criteria change that changes the effective birth prevalence affects the entirety of any birth cohort where the birth year is greater than or equal to the year the change took effect. For birth years before the year of criteria change, a change in criteria affects *EBP* and the size of the risk set *R* starting at the diagnostic year the change took effect. Generally, diagnostic criteria should be given in published documents specifying the effective dates of new or revised criteria.

Let {*CFcy*} be the set of **criteria factors** that induce a multiplicative effect on *EBP* due to criteria changes that occurred at criteria years {*cy*}. $CF_0$, the value in effect before the first *DY*, equals 1. $BP_{BY}$ is the birth prevalence of cohort *BY* before the effect of any criteria changes. For each cohort *BY*, $EBP_{BY,A}$ at age *A* is

$$EBP_{BY,A} = BP_{BY}\prod_{cy \leq (BY+A)}CF_{cy} \tag{8}$$

The combination of $BP_{BY}$ and the effects of all $\{CF_{cy}|cy \leq BY + A\}$ determines the *EBP* at age *A* of cohort *BY*.

For a given *BY* and increasing *A*, *BY* + *A* crossing any criteria year *cy* causes a step-change in *EBP*. Using a general formulation supporting increasing eligibility $E_A$ based on Eq (3) we obtain the following. $A = 0$, $R_{BY,0} = E_0 EBP_{BY,0}$ and $D_{BY,0} = E_0 EBP_{BY,0} PD_{BY,0}$. For $A \geq 1$,

$$R_{BY,A} = R_{BY,A-1} - D_{BY,A-1} + E_A(EBP_{BY,A} - EBP_{BY,A-1}) + (E_A - E_{A-1})EBP_{BY,A}$$

and

$$D_{BY,A} = [R_{BY,A-1} - D_{BY,A-1} + E_A(EBP_{BY,A} - EBP_{BY,A-1}) + (E_A - E_{A-1})EBP_{BY,A}]PD_{DY} \tag{9}$$

The term $EBP_{BY,A} - EBP_{BY,A-1}$ represents the change in *EBP* when *BY* + *A* crosses one of {*cy*}. As each *CFcy* takes effect at *cy* = *DY* = *BY* + *A*, the newly effective *CFcy* changes $EBP_{BY,A}$ and $R_{BY,A}$ in all *BY* cohorts where *cy* corresponds to an age *A* in the range of ages included in the analysis. These changes in $R_{BY,A}$ affect the rates of initial diagnoses *D*. For cohorts born after *cy*, *CFcy* applies to all ages.

The parameters of $BP_{BY}$ quantify the birth prevalence controlled for diagnostic criteria changes, which are represented by {*CFcy*}. In other words, $BP_{BY}$ is the effective birth prevalence that would have occurred if the initial criteria had been applied at all diagnostic years included in the study.

To estimate the parameters, use a software model of Eq (9) with optimization software as described above.

## Potential violations of assumptions

Suppose a dataset represents a non-homogeneous set of cases with different effective values of *PD* applying to different unidentified subgroups at the same *DY*. That would violate the assumption that *PD* applies equally to all eligible undiagnosed cases at any given *DY*. Cases may have differing degrees of symptom severity, and more severe symptoms may result in earlier diagnosis [28, 41–43], implying greater values of *PD*. Fig 3 illustrates this situation. (The figure illustrates constant values of *PD* vs. age purely for clarity, not as an assumption nor a limitation.) If the data represent a combination of unidentified subgroups with different values of *PD*, the distribution of diagnoses is a sum of distributions with different values of *PD*. Such

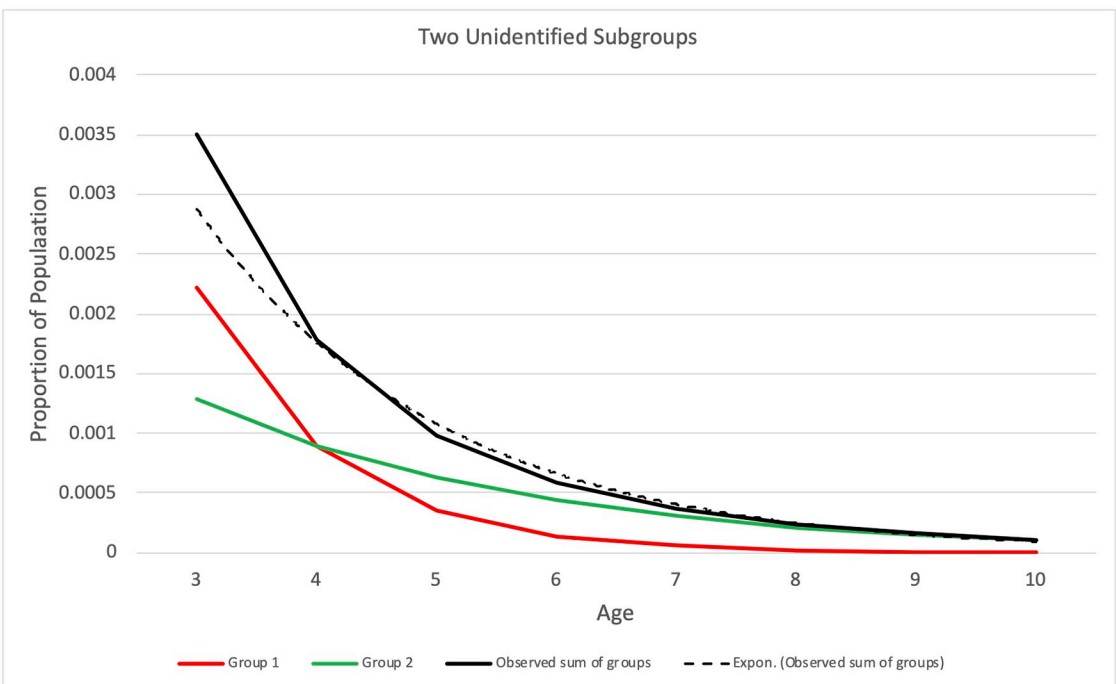

**Fig 3. Example where observed diagnosis rates represent two unidentified subgroups with different values of probability of diagnosis *PD*.** The red and green lines represent rates of diagnosis of the two subgroups. Group 1 (red line) has greater *PD* than group 2 (green line). The solid black line shows the aggregate diagnosis rates. The dotted line shows the exponential fit to the aggregate diagnosis rates. The age of complete eligibility *AE* = 3 in this example.

a sum may impair fit with a model that assumes homogeneous *PD*. For data where $E = 1$ for $A \geq AE$ (plateau $E_A$), slightly increasing the assumed value $AE^*$ of the age of complete eligibility *AE* used in estimation may mitigate such estimation errors, as shown in the simulation study below. If subgroup data are available representing groups according to symptom severity or other characteristics that may affect *PD*, stratified estimation using subgroups may avoid the issue of unidentified non-homogeneous subgroups. If one suspects non-homogeneity tied to geographic location, such as differences in diagnostic practices or health disparities, stratification by geographic location can elucidate such differences. The same applies to suspected non-homogeneity tied to characteristics such as race, ethnicity or socio-economic status.

If there is a difference in *BP* between the populations migrating into and out of the study region, that would change the *BP* of individual cohorts over time. If some cases of the disorder are caused by exposures after birth, and those exposures vary by year, that would also change *BP* over time. Either effect would violate the assumption of constant birth prevalence within each cohort. If post-natal exposures caused increased *BP* before age *AE*, then estimates from analyses that use empirical values of *D* for ages less than *AE* would represent values of *BP* that include the post-natal effect.

It is theoretically possible for *PD* to have different effective values for cases of different ages with the same symptom severity at the same *DY*. Such an effect would represent an age bias in *PD* independent of symptom severity. Investigators can examine this possibility by adding an age term to *PD* in the model and estimating its parameters. One form of a potential age bias in *PD* would be age-specific screening for the disorder. In 2006, the American Academy of Pediatrics [46] recommended screening tests for developmental disorders be administered at 9-, 18- and 30- (or 24-) month visits. In 2011, Al-Qabandi [47] concluded that autism screening

programs were generally ineffective. Screening in the USA starting approximately 2006 might plausibly have increased the effective *PD* for ages 0 to 2 or 3. An analysis could test for a screening-related increase in *PD* by including a variable for increased *PD* at ages 0 through 3 starting in 2006. For models that use empirical rates of diagnoses for ages less than *AE*, the probability of diagnosis at those ages is not part of the model and does not affect parameter estimates. Analysis could include a parameter specifying increased *PD* at age *AE*, such as age 3. Alternatively, analysis could specify *AE* = 4 such that the probability of diagnosis is estimated for ages $\geq$ 4, thereby avoiding any potential bias associated with increased probability of diagnosis specific to age 3.

If some in-migrating cases were diagnosed before in-migration and their subsequent re-diagnoses in the study region were labeled as initial diagnoses, that would violate the assumption of truly first diagnoses. Such an effect would be most evident at greater ages after the diagnosis of most cases. Bounding the maximum age studied *M* to a modest value, sufficient to capture most initial diagnoses, can minimize any resulting bias.

For datasets where diagnosis follows best practices using gold-standard criteria, the lack of false positives may be a fair assumption, but they might occur nevertheless. It may be difficult to discover any false positives produced from best practice diagnoses. Where diagnosis uses a less precise process, false positives might occur. For example, diagnosticians might produce a diagnosis of individuals who do not meet formal diagnostic criteria, perhaps under pressure from the patient or parents, or to facilitate services for the individual. Meyers [12] noted the risk of false positives in record-based approaches in contrast to clinical diagnoses. Lord [48] stated that what are sometimes called autism spectrum disorder symptoms that appear at later ages are not always related to autism spectrum disorder. Suppose the pool of individuals subject to false-positive diagnoses is substantially larger than the birth prevalence of true cases. In that case, false-positive diagnoses deplete the pool of such individuals slowly relative to the survival function of true cases. That may result in an approximately uniform age distribution of false-positive diagnoses, causing a nearly constant additive offset to the rates of diagnoses across ages. True case diagnoses should be more common at younger ages, and less common as the risk pool is depleted, so false positives may be more evident at older ages. Setting the maximum age of follow-up *M* to a moderate value, such as age 8 or 10 for autism, may help reduce the impact of false positives on estimates while enabling accurate estimation.

The assumption that case status is binary may not be completely valid, at least in the case of autism or ASD. Different diagnostic assessment tools, assessments by different professionals, and applying different cut-off thresholds within a tool can produce somewhat different results [49].

The formulation that birth prevalence equals the cumulative incidence to age *M* plus the right-censored portion assumes that any difference in competing risks between cases and non-cases in the age range analyzed is small enough to be ignored. This assumption is consistent with Hansen [26]. In contrast, if the rate of deaths of cases before initial diagnosis exceeds the rate in the cohort's overall population at the same ages, that excess would constitute a competing risk and would reduce the estimated birth prevalence accordingly.

## Model fit

To ensure robust conclusions, investigators should test the model fit to ascertain both model correctness and parameter estimation accuracy. The model fits well if summary measures of the error are small and individual point errors are unsystematic and small [50]. One can examine the fit both graphically and numerically. Plots of $D_{BY,A}$ vs. $\widehat{D_{BY,A}}$ at all ages for individual cohorts and separately at single ages across all cohorts can illuminate any issues with fit, which

might occur at only some cohorts or ages. Visualization of the model vs. data can expose aspects of the data that might not fit well in a model with few parameters, suggesting a higher-order model or semi-parametric specifications.

Suppose the model uses an assumed age of complete eligibility $AE^*$ that differs from the true value of $AE$ underlying the data. That mismatch may impair model fit. The simulation study section shows that setting $AE^* < AE$ can result in estimation errors. However, setting $AE^* > AE$ tends not to impair model fit and may improve it in the case of non-homogeneous subgroups; see Fig 3. The presence of non-homogeneous subgroups may be evident from examining model fit.

The chi-square test statistic is a numerical approach to assess absolute model fit. Investigators can apply it to the overall model, individual cohorts, and single ages across cohorts. The p-value associated with the chi-square statistic uses observed and expected count values rather than proportions. The p-value incorporates the effect of the number of parameters in the model via the degrees of freedom.

## Simulation study

This section presents a simulation study to measure the performance and accuracy of the TTEPE method, following the recommendations in Morris [51]. Simulation studies evaluate performance by generating pseudo-random synthetic data according to known parameters over a broad range of parameter values, estimating parameters from that data using the method under evaluation, and comparing the estimates to the known true parameter values. Simulation studies can detect potential problems such as ambiguity or bias [51].

The simulation study described here uses a model with first-order exponential forms of birth prevalence $BP$ and probability of diagnosis $PD$. It uses plateau eligibility $E$ with the age of complete eligibility $AE = 3$. There are 20 successive cohorts, and the last age of follow-up is $M = 10$. The parameters of primary interest are the exponential coefficients of birth prevalence $\beta_{BP}$ and probability of diagnosis $\beta_{PD}$. The study tested six pairs of values of $\beta_{BP}$ and $\beta_{PD}$, each ranging from 0 to 0.1 in steps of 0.02, where each pair sums to 0.1. In one parameter set, $BP$ increases as $e^{0.1 \times BY}$ (10.5% per year) and $PD$ is constant. In another parameter set, $BP$ is constant and $PD$ increases as $e^{0.1 \times DY}$ (10.5% per year). The other four parameter sets represent various rates of change of both variables. In all cases, $BP = 0.01$ at the final $BY$ and $PD = 0.25$ at the final $DY$. These simulations assume the investigators chose the correct value of the age of complete eligibility $AE = 3$, following the plateau $E_A$ model, from either knowledge of the disorder or estimation of the eligibility function $E_A$. The study synthesized each data model in two ways: real-valued proportions without sampling, and a Monte Carlo model with binomial random sample generation. The use of real-valued proportions tests the estimation method's accuracy (bias) in the absence of sampling effects in the data. It is logically equivalent to testing estimation bias in an infinitely large population. Monte Carlo simulation generated data sets using binomial sampling of case counts for each birth cohort and counts of initial diagnoses at each age within each cohort, with 1000 iterations of random data set generation for each parameter set. The population of each synthetic cohort is a constant of 500,000. TTEPE analysis estimated the parameters for each iteration. The results show the parameter estimation bias and model standard error (SE) for each parameter set over all iterations. The study estimated the parameters using the method described above, implemented using the Python SciPy curve_fit() function. The software that performed this study is publicly available [45].

Table 1 shows results using real-valued proportions without sampling, which isolates the estimation process from random sampling variations. It shows the bias in estimating each of the four model parameters for each of the six synthesis parameter sets. The biases are minimal

**Table 1. Simulation results of parameter optimization using real-valued proportions with no sampling.**

| True Parameters | | Bias $\widehat{BP}$ at final $BY$ | Bias $\widehat{\beta}_{BP}$ | Bias $\widehat{PD}$ at final $DY$ | Bias $\widehat{\beta}_{PD}$ |
|---|---|---|---|---|---|
| $\beta_{BP}$ | $\beta_{PD}$ | | | | |
| 0.1 | 0 | 5.9E-12 | 8.9E-11 | -5.5E-10 | -1.8E-10 |
| 0.08 | 0.02 | 0 | 0 | -2.8E-17 | -1.4E-17 |
| 0.06 | 0.04 | -1.7E-18 | 0 | 1.1E-16 | 1.4E-17 |
| 0.04 | 0.06 | 3.5E-18 | 1.4E-17 | -5.6E-17 | -6.9E-18 |
| 0.02 | 0.08 | 0 | 3.5E-18 | 5.6E-17 | -1.4E-17 |
| 0 | 0.1 | -4.7E-11 | -6.6E-10 | 2.2E-9 | 7.7E-10 |

$BP$ = 0.01 at the final $BY$, $PD$ = 0.25 at the final $DY$, $AE^* = AE = 3$, $M = 10$, and there are 20 successive cohorts.

$\beta_{BP}$, $\beta_{BD}$ are exponential coefficients for birth prevalence and probability of diagnosis, respectively.

$BP$, birth prevalence; $BY$, birth year; $DY$, diagnostic year.

and may be caused by finite precision arithmetic in the computer. The greatest bias magnitude in $\widehat{\beta}_{BP}$ occurs with $\beta_{BP} = 0$ and $\beta_{PD} = 0.1$ and is on the order of $10^{-10}$. This result shows that the parameter estimation is extremely accurate in the absence of sampling effects.

Table 2 shows the Monte Carlo analysis of the same parameter sets where the data use binomial sampling. It shows the bias and model standard error (SE) of each parameter for each parameter set. The bias of the primary parameter $\widehat{\beta}_{BP}$ remains small, on the order of $10^{-5}$ or $10^{-6}$. The SE is relevant when there is sampling, and it shows the effect of sampling, in contrast to Table 1.

Table 3 gives results where estimation uses an assumed value $AE^*$ of the age of complete eligibility $AE$ that, in some cases, does not match the true value of $AE = 3$ used to synthesize the data. Synthesis uses one homogeneous group with consistent $PD$ at each value of $DY$. Estimation used various assumed values of $AE^*$ to test the effect of the choice of $AE^*$. Estimation using $AE^* = 2$ results in substantial estimation errors and model misfit that is obvious from plots of data vs. model (not shown). Estimation using $AE^* = 3$, $AE^* = 4$, or $AE^* = 5$ produces accurate results, with slightly more error where $AE^* = 5$. Plots show that the model fits well in all three cases (not shown). Thus, the choice of $AE^*$ is not critical as long as $AE^* \geq AE$. These

**Table 2. Simulation results of parameter optimization using Monte Carlo binomial sampling, 1000 iterations.**

| True Parameters | | $\widehat{BP}$ at final $BY$ | | $\widehat{\beta}_{BP}$ | | $\widehat{PD}$ at final $DY$ | | $\widehat{\beta}_{PD}$ | |
|---|---|---|---|---|---|---|---|---|---|
| $\beta_{BP}$ | $\beta_{PD}$ | Bias | SE | Bias | SE | Bias | SE | Bias | SE |
| 0.1 | 0 | -2.0E-6 | 1.0E-4 | -2.0E-5 | 0.0013 | 3.3E-5 | 0.0070 | -5.4E-6 | 0.0019 |
| 0.08 | 0.02 | -2.6E-6 | 1.1E-4 | -3.2E-5 | 0.0012 | 1.5E-4 | 0.0072 | 4.4E-5 | 0.0019 |
| 0.06 | 0.04 | 7.8E-6 | 1.2E-4 | 2.7E-5 | 0.0013 | -4.4E-4 | 0.0079 | -1.2E-4 | 0.0021 |
| 0.04 | 0.06 | 1.3E-5 | 1.5E-4 | 6.5E-5 | 0.0015 | -5.8E-4 | 0.0085 | -1.6E-4 | 0.0022 |
| 0.02 | 0.08 | -2.0E-6 | 1.6E-4 | -9.8E-6 | 0.0016 | 4.5E-4 | 0.0086 | 9.4E-5 | 0.0023 |
| 0 | 0.1 | 4.5E-6 | 1.8E-4 | 7.1E-6 | 0.0017 | 2.0E-4 | 0.0094 | 2.7E-5 | 0.0023 |

$BP$ = 0.01 at the final $BY$, $PD$ = 0.25 at the final $DY$, $AE^* = AE = 3$, $M = 10$, and there are 20 successive cohorts.

Population of each cohort = 500,000.

$\beta_{BP}$, $\beta_{DP}$ are exponential coefficients for birth prevalence and probability of diagnosis, respectively.

$BP$, birth prevalence; $BY$, birth year; $DY$, diagnostic year.

**Table 3. Comparison of the effect of the choice of assumed $AE^*$ vs. true value of $AE = 3$, with one homogeneous group of cases.**

| $AE^*$ used in estimation | Bias $\widehat{BP}$ at final BY | Bias $\widehat{\beta_{BP}}$ | Bias $\widehat{PD}$ at final DY | Bias $\widehat{\beta_{PD}}$ |
|---|---|---|---|---|
| 2 | 0.002 | -0.019 | -0.0096 | 0.036 |
| 3 | 5.9E-12 | 8.9E-11 | -5.5E-10 | -1.8E-10 |
| 4 | -4.4E-12 | -6.6E-11 | 4.8E-10 | 1.64E-10 |
| 5 | 1.5E-11 | 2.2E-10 | -1.85E-9 | -7.18E-10 |

$AE$, age of complete eligibility. True values: $\beta_{BP} = 0.1$, $\beta_{PD} = 0$, $P = 0.01$ at the final $BY$, $PD = 0.25$ at the final $DY$, $AE = 3$. Maximum age $M = 10$. Twenty successive cohorts. Probability of diagnosis $PD$ is consistent across cases at each $DY$. Simulation uses real values, no random sampling.

**Table 4. Comparison of the effect of the choice of assumed $AE^*$ vs. true value of $AE = 3$, with two unidentified subgroups with different values of probability of diagnosis, mismatched to analysis.**

| $AE^*$ used in estimation | Bias $\widehat{BP}$ at final $BY$ | Bias $\widehat{\beta_{BP}}$ | Bias $\widehat{PD}$ at final $DY$ | Bias $\widehat{\beta_{PD}}$ |
|---|---|---|---|---|
| 3 | -0.00043 | 0.001 | 0.0018 | -0.002 |
| 4 | -0.00038 | 0.00061 | -0.004 | -0.0016 |
| 5 | -0.00033 | 0.00035 | -0.0097 | -0.0011 |

$AE$, age of complete eligibility. True values: $\beta_{BP} = 0.1$, $\beta_{PD} = 0$, $P = 0.01$ at the final $BY$, $PD = 0.25$ at the final $DY$, $AE = 3$. Two equal-sized groups of cases where one group's probability of diagnosis $PD$ is twice that of the other, while the estimation assumes one homogeneous group. Maximum age $M = 10$. Twenty successive cohorts. Simulation uses real values, no sampling.

data use real-valued proportions rather than binomial sampling to avoid confusion of model mismatch with sampling effects.

Table 4 shows results with an intentional mismatch between estimation that assumes one homogeneous group and data that comprises two unlabeled subgroups with different values of $PD$, illustrated in Fig 3. Fig 3 shows the visible error of the exponential fit to the data at age = 3 and a good fit for age > 3. In this synthetic dataset, the two subgroups are of equal size, and the true value of $PD$ in one group is twice that of the other. This information is unknown to the estimation, and the data does not indicate subgroup size or membership. In the worst case, estimation uses $AE^* = AE = 3$, and the $\widehat{\beta_{BP}}$ bias is 0.001, which is 1% of the actual value of 0.1. This error is due to the model misfit at age three resulting from subgroups having different values of $PD$, illustrated in Fig 3, and the estimation does not account for the inconsistent values of $PD$. When using $AE^* = 4$ or $AE^* = 5$, the biases in $\widehat{\beta_{BP}}$, final $\widehat{BP}$, and $\widehat{\beta_{PD}}$ are reduced while the bias in final $\widehat{PD}$ is increased, all by small amounts.

## Discussion

The TTEPE method solves the previously intractable problem of accurately estimating birth prevalence properly adjusted for diagnostic factors and changes in diagnostic criteria. It addresses the requirements stated in the introduction. The simulation study shows that TTEPE produces accurate estimates of the true parameter values across a broad range of values and in the presence of random sampling effects. This performance implies that TTEPE is suitable for estimating birth prevalence.

TTEPE is novel. Apparently, no previous papers use time-to-event or survival analysis to estimate birth prevalence. In typical survival or time-to-event analysis, including Cox

proportional hazards analysis [52], the initial size of the risk set is assumed to have a known value, such as an entire population, group, or sample, in contrast to the problem addressed here. For models where the risk set at all ages consists of the entire population without sub-tracting diagnosed cases, the estimated hazard function of time $\widehat{PD_t} = D_t/R_t$ is simply the population-based rate of diagnoses $D$ at time $t$ [19, 28, 42]. Such models to not apply a survival process to the size $R$ of the risk set; see above and Kalbfleisch [44]. For methods using the entire population as the initial risk set, if the disorder is rare, it makes little difference whether the risk set at subsequent ages is the entire population or the undiagnosed portion since the size of the risk set decreases only slightly as cases are diagnosed. Some papers use survival analysis methods to estimate other measures, such as the median age at first diagnosis [28, 42, 43]. That usage is not directly relevant to this paper. Szklo [1] describes the use of survival analysis to calculate cumulative incidence. That approach does not apply here because it depends on two assumptions that are not valid for the present problem. One is a lack of a birth cohort trend, and the other is independence between censoring and survival, which conflicts with a varying probability of diagnosis.

TTEPE avoids the problems of methods that assume, ignore, or inappropriately estimate the age distribution, including age-period-cohort methods. It achieves this by directly modeling the age distribution of first diagnoses via a non-linear function derived from first principles. A figure in [S1 Text] illustrates why fitting the age distribution of diagnoses via a survival model enables identifying correct parameter values. As noted in the background section, some analytic methods adjust for variables on the causal path, which leads to biased estimates. The TTEPE method likewise avoids that problem.

This paper states the assumptions that underlie TTEPE analysis and discusses potential violations of those assumptions. The DAG of Fig 2 illustrates the assumed causal paths from birth year, diagnostic year, and age, including the effect of the set of time-varying diagnostic factors on the probability of diagnosis and the effect of changes in diagnostic criteria on effective birth prevalence. The DAG and associated analysis appear to cover all plausible mechanisms to explain observed trends in rates of initial diagnoses.

The simulation study presented above shows that TTEPE produces estimates with minimal bias and strong power to resolve parameter values from input data with sampling effects. The results in Table 2 show a magnitude of bias of the birth prevalence exponential coefficient $\widehat{\beta_{BP}}$ not exceeding 6.5 x $10^{-5}$, i.e., 0.0065% per year. The model standard error (SE) of $\widehat{\beta_{BP}}$ ranges from 0.0012 to 0.0017, where the true $\beta_{BP}$ ranges from 0 to 0.1. Using the largest observed SE and 1.96 x SE as the 95% confidence interval threshold, the simulated model can resolve differences in $\beta_{BP}$ of 0.0033, i.e., 0.33% per year. Investigators can expect similar performance for real-world datasets that meet the baseline assumptions and have characteristics comparable to the simulated data. The population size and birth prevalence affect the numbers of incident diagnoses and hence the SE. Note that there are 20 cohorts and 11 ages (0 through 10) in the simulation study, so there are 220 data points. Each data point is an independent binomial random sample. The analysis estimates four parameters that define the curves that fit the data. The relatively large number of independent data points and the small number of model parameters help produce the small bias and model SE. If each cohort's population or the birth prevalence had been substantially smaller, or the number of parameters had been greater, we would expect the SE and possibly the bias to be larger. These could occur with small geographic regions or rare disorders, or higher-order or semi-parametric models, respectively.

It is possible but challenging to prove mathematically that a model is identifiable [38], and we have not done that. We have not found any evidence that the models described here are unidentified. Investigators can construct a wide variety of TTEPE models with various

parametric or non-parametric forms of the variables described here and potentially additional variables. As such, it is possible to construct models with collinearities and resulting non-identification. Simultaneously estimating the parameters across multiple birth cohorts helps discriminate between parameter sets that might interact within one cohort. As with any regression, investigators using TTEPE should ensure there are enough data points to estimate parameters with sufficient precision.

Logically, the eligibility function $E_A$ should be an attribute of the disorder under study. Specifically regarding autism or ASD, the literature shows that cases begin to show predictive symptoms well before age three, and some are diagnosed at age two [49]. According to some but not all diagnostic criteria for autism and ASD, symptoms must be present by age three [29]. There is evidence that some cases with milder symptoms who do not meet diagnostic criteria at age three do meet criteria at a later age [53]. Most of those diagnosed at an early age develop more severe symptoms over time [54]. There is also evidence that some individuals who meet or appear to meet case criteria before three years of age no longer meet criteria at some later age [55]. These phenomena are consistent with the discussion above of non-homogeneous subgroups with different effective values of the probability of diagnosis. The phenomenon of late development of diagnosable symptoms may be worthwhile to investigate, particularly with datasets representing initial diagnoses over a broad range of ages. One way to model that phenomenon is the eligibility function continuing to increase over many years of age. It may also be worthwhile to investigate the proportion of cases diagnosed after childhood that do not meet formal criteria, i.e., false positives. As Tables 3 and 4 show, some errors in estimating the eligibility function and erroneously assuming the homogeneity of the severity of cases have only a minor effect on parameter estimates when the assumed age of complete eligibility $AE^*$ is chosen carefully.

Investigators can use TTEPE to answer important questions beyond the overall trend in birth prevalence. For example, where datasets include appropriate covariates, stratified analysis can determine whether observed trends in the rates of diagnoses in specified subgroups are due to actual changes in birth prevalence or diagnostic factors. Example covariates include sex, race, ethnicity, socio-economic status, geographic region, parental age, environmental exposure, genetic profile, and other factors of interest.

Investigators may use domain knowledge to inform specialized analyses. For example, they may incorporate knowledge of mortality rates and standardized mortality ratios, rates of recovery from the condition before diagnosis, or the characteristics of migration in and out of the study region.

It may be feasible to extend TTEPE to disorders, diseases and conditions where the time scale starts at some event other than birth. For example, the time origin might be the time of completion of a sufficient cause. Various outcomes may serve as events of interest. It is important to ensure that the eligibility function with respect to the time origin is consistent across cohorts.

## Supporting information

**S1 Text. TTEPE additional analyses and explanations.** Contains figures.
(PDF)

## Acknowledgments

The author thanks Dr. Lu Tian for his expert advice on survival analysis methods; Dr. Lorene Nelson and Dr. Kristin Sainani for guidance on my thesis, which was the genesis of this project

and for comments on this paper; Dr. Michael Sigman, Dr. Larry Tang and Dr. Walter Zahor-odny for their thoughtful reviews of the paper; the Stanford Biomedical Data Science team for their project reviews and insightful comments; and Anne Goff for invaluable editorial assistance.

## Author Contributions

**Conceptualization:** Alexander G. MacInnis.

**Formal analysis:** Alexander G. MacInnis.

**Investigation:** Alexander G. MacInnis.

**Methodology:** Alexander G. MacInnis.

**Project administration:** Alexander G. MacInnis.

**Software:** Alexander G. MacInnis.

**Validation:** Alexander G. MacInnis.

**Visualization:** Alexander G. MacInnis.

**Writing – original draft:** Alexander G. MacInnis.

**Writing – review & editing:** Alexander G. MacInnis.

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
