## [Decision Letter · Decision Letter 0]

16 Feb 2021

PONE-D-20-24616

Time-to-event estimation of birth year prevalence trends: a method for investigating etiology of childhood disorders including autism

PLOS ONE

Dear Dr. MacInnis,

Thank you for submitting your manuscript to PLOS ONE. After careful consideration, we feel that it has merit but does not fully meet PLOS ONE’s publication criteria as it currently stands. Therefore, we invite you to submit a revised version of the manuscript that addresses the points raised during the review process.

We look forward to receiving your revised manuscript.

Kind regards,

Jiayin Wang, Ph.D.

Academic Editor

PLOS ONE

Journal Requirements:

Reviewers' comments:

Reviewer's Responses to Questions

**Comments to the Author**

1. Is the manuscript technically sound, and do the data support the conclusions?

Reviewer #1: Partly

Reviewer #2: Yes

2. Has the statistical analysis been performed appropriately and rigorously? 

Reviewer #1: I Don't Know

Reviewer #2: I Don't Know

3. Have the authors made all data underlying the findings in their manuscript fully available?

Reviewer #1: Yes

Reviewer #2: Yes

4. Is the manuscript presented in an intelligible fashion and written in standard English?

Reviewer #1: No

Reviewer #2: Yes

5. Review Comments to the Author

Reviewer #1: The paper addresses an important topic—i.e., changes over time in incidence and prevalence of chronic disorders, such as autism. By making us of full distributions of age-of-onset data, the proposed approach may well offer an important contribution.

Unfortunately, I find the paper as written to be difficult to understand. For example:

1. Many technical terms are used but only few are defined. As the author recognizes on p 3, technical terms, such as “incidence”, are often used in different ways. Thus, it is important to be clear about how all terms are used. Unfortunately, I often found definitions to be unclear. Examples include:

a. P3: “incidence” – “the rate of new cases” (p 3)—to me, this suggests that incidence should include reference to time, e.g., annual inference.

b. P3: “Prevalence” – “proportion of a defined population with the disorder at a defined time.” Does “time” here refer to year or age or something else?

c. P3: “birth year cohort prevalence”—birth year is not enough to define a population; one also needs to state the current age of that cohort

d. P3: “disorder incidence” is “indistinguishable from birth prevalence”. I assume birth prevalence = birth year cohort prevalence. I think this can only be indistinguishable from disorder incidence if incidence is calculated over a time period equal to the current age of the cohort. But I’m not sure, because the use of terms is confusing me. For example, on p 5, the author states that CDC reports of 8 year-olds reflect “birth year prevalence.” This may be the same as the rate of new cases over the 1st 8 years of life, but I seldom see 8-year incidence reported. Note that some studies discuss “8 year probability” or “lifetime probability” of disorder to address this.

e. P3: “rate of incident diagnoses rather than true incidence”—again, I think I know what the author means by diagnoses vs “true” incidence, but I can’t be sure unless they are defined.

f. P 5: “cumulative incidence” is mentioned. How is this different than “birth year prevalence” assessed at a certain age?

g. P 20 “case prevalence” is mentioned. Do cases represent “true disorders” or only those who are “eligible” based on current symptoms?

2. The paper is quite long and includes many examples that do not help to advance the argument. For example:

a. there are numerous simulations, and I do not understand how Figure 1 helps

b. The new method is not described in detail until p 16. There are many paragraphs with 1 or 2 sentences that seem to make an ancillary point (e.g., re: occum’s razor on p 14)

c. The DAG is not well-specified. For example, the author states that “diagnosable symptoms and diagnostic pressure together produce each initial diagnosis.” This sounds like an interaction to me, yet it is presented as if these were independent effects

d. The simulations are difficult to follow (at best), so I do not find them persuasive.

At its core, the paper attempts to answer the question on p 4, “If reported birth prevalence increased over time, was this caused by changes in actual birth prevalence, the probability of diagnosing cases, diagnostic criteria affecting prevalence, or some combination of these factors?” As the author states later in the paper (p 7), many have discussed challenges to disentangling effects attributable to age, period, and cohort—i.e., quoting O’Brien who wrote, “there is no way to decide except by making an assumption about the relationship between these three variables.” In response, the author proposes a method based on several assumptions, yet notes often that it “correctly adjusts for changes in the set of diagnostic factors and diagnostic criteria.” The word “correctly” is used several times to describe the new method, but I don’t understand how this is appropriate when it is based on assumptions. For example, it is fine to assume that “diagnostic pressure” is equal across ages, but is this correct given that screening is only recommended at certain ages?

A more parsimonious argument would be helpful. As the author begins the discussion, “Readers may suspect that the estimates are unidentified, i.e., not unique, due to possible interaction between age, diagnostic year, and birth year, such that estimates may be biased even if the model fit is excellent.” It is not enough to state that “TTEPE avoids that problem and produces uniquely identified estimates.” As a reader who lacks the expertise to process the equations in detail (as I suspect many readers will be), a clear description is essential. For example, it is assumed that the eligibility function is consistent across cohorts (p 22) and that it is an “attribute of the disorder under study”—does this imply that the entire function is assumed, or just its functional form? How does TTEPE avoid identification problems when so many parameters (eligibility, incidence, diagnostic pressure, etc) are being estimated?

Clear and cogent answers to questions such as these will facilitate review of the merits of the proposed method and—assuming it is indeed correct—promote is wider dissemination.

Reviewer #2: This paper addresses a very important topic – monitoring the prevalence of ASD over time given changes in diagnostic practices. The emphasis, by virtue of including parameters of rates of initial diagnosis by diagnostic year and age for each birth cohort and using simulation studies to understand the benefits and limitations of the method is a novel and useful approach. Prior to presenting my review, I wish to disclose that while I have written papers on the prevalence of childhood disorders, have used survival analyses to model prevalence, and have been very interested in following ASD prevalence papers, I do not have the mathematical knowledge to evaluate this element of the modeling. This said, I believe that the author has done a great job of reviewing the challenges and controversies in this area, making a strong case for the need for new methods to disentangle potential changes in prevalence of ASD over time from diagnostic practice factors such as changing diagnostic criteria and greater outreach with screening. The presentation of the method is extremely clear.

This approach builds on survival analysis models in recognizing that risk for diagnosis continues through the lifespan as well as incorporating changing clinical practices in relation to diagnosing ASD and IDD.

Distinguishing incidence and prevalence may be more important when conditions are not enduring or are episodic such as depression.

I appreciate that the author highlights the potential for lack of homogeneity in diagnostic pressure within cohorts. Unfortunately, this is likely to be the case as regional variation in practice continues to be quite variable and health disparities may also contribute to heterogeneity in practice within cohorts.

It might also be useful to recognize that ASD diagnoses generally begin to be possible in the second year of life with many cases emerging in the third year of life (see Zwaigenbaum et al., Ozonoff et al., who each follow diagnoses within infant sibling samples) and that there is greater instability in early diagnoses (i.e., under age 2) than when diagnoses are made in later preschool (e.g., Giserman Kiss & Carter, 2019).

The clarity of the writing is a major strength of this paper. My only suggestion is to incorporate more findings that impact the timing and heterogeneity of diagnoses into the discussion.

6. PLOS authors have the option to publish the peer review history of their article (what does this mean?). If published, this will include your full peer review and any attached files.

Reviewer #1: No

Reviewer #2: No

---

## [Author Response · Author response to Decision Letter 0]

11 Mar 2021

Please see the Response to Reviewers file which I have uploaded as part of this revised submission.

---

## [Decision Letter · Decision Letter 1]

18 Jun 2021

PONE-D-20-24616R1

Time-to-event estimation of birth year prevalence trends: a method to enable investigating the etiology of childhood disorders including autism

PLOS ONE

Dear Dr. MacInnis,

Thank you for submitting your manuscript to PLOS ONE. After careful consideration, we feel that it has merit but does not fully meet PLOS ONE’s publication criteria as it currently stands. Therefore, we invite you to submit a revised version of the manuscript that addresses the points raised during the review process.

We look forward to receiving your revised manuscript.

Kind regards,

Jiayin Wang, Ph.D.

Academic Editor

PLOS ONE

Journal Requirements:

Additional Editor Comments (if provided):

Please carefully consider the comments from reviewer4.

Reviewers' comments:

Reviewer's Responses to Questions

**Comments to the Author**

1. If the authors have adequately addressed your comments raised in a previous round of review and you feel that this manuscript is now acceptable for publication, you may indicate that here to bypass the “Comments to the Author” section, enter your conflict of interest statement in the “Confidential to Editor” section, and submit your "Accept" recommendation.

Reviewer #3: (No Response)

Reviewer #4: (No Response)

2. Is the manuscript technically sound, and do the data support the conclusions?

Reviewer #3: (No Response)

Reviewer #4: Partly

3. Has the statistical analysis been performed appropriately and rigorously? 

Reviewer #3: (No Response)

Reviewer #4: I Don't Know

4. Have the authors made all data underlying the findings in their manuscript fully available?

Reviewer #3: (No Response)

Reviewer #4: Yes

5. Is the manuscript presented in an intelligible fashion and written in standard English?

Reviewer #3: (No Response)

Reviewer #4: Yes

6. Review Comments to the Author

Reviewer #3: (No Response)

Reviewer #4: The paper addresses an important topic—birth year prevalence estimation adjusted for time-varying diagnostic factors and diagnostic criteria. By making use of the time-to-event survival method to model the causal paths from birth year, diagnostic year, and age at diagnosis, the proposed approach may well offer an important contribution. Unfortunately, even after the revision, this paper is still too long and difficult for the reader to understand.

For example:

1. There is a logical confusion between the description of “incidence” and “prevalence” in Introduction part. As the author states on p3, “Incidence is different from prevalence” and “Prevalence is rarely of direct interest in studying etiology”, does it seem unreasonable to use birth-year cohort prevalence rather than incidence in this paper? The definition and distinction between “incidence” and “prevalence” and why "prevalence" was chosen still needs further elaboration and clarification.

2. The article is too long and some parts do not need to be separate paragraphs. The Literature Review section is overly long and not very meaningful to the model presented in this article. The description of the strengths and weaknesses of the available methods needs to be further streamlined and summarized rather than simply listed. The Overview section needs to be further refined and integrated into the Introduction section.

3. The Background part in Method of time-to-event prevalence estimation (TTEPE) appears to be a duplicate of Introduction and Literature Review.

4. Ambiguity in estimation part does not seem necessary, and a more parsimonious argument would be helpful. A quick entry into the model specification would have been better for the reader to understand the point of the article.

This paper uses a time-varying survival analysis model to portray birth year prevalence trend is reasonable and novel. While, the structure of the article still needs to be reorganized, and the key points needs to be sharpened rather than simply narrated.

7. PLOS authors have the option to publish the peer review history of their article (what does this mean?). If published, this will include your full peer review and any attached files.

Reviewer #3: No

Reviewer #4: **Yes: **Yixuan Wang

---

## [Author Response · Author response to Decision Letter 1]

30 Sep 2021

Please see the attached Response to Reviewers.

---

## [Decision Letter · Decision Letter 2]

17 Nov 2021

Time-to-event estimation of birth prevalence trends: a method to enable investigating the etiology of childhood disorders including autism

PONE-D-20-24616R2

Dear Dr. MacInnis,

We’re pleased to inform you that your manuscript has been judged scientifically suitable for publication and will be formally accepted for publication once it meets all outstanding technical requirements.

Kind regards,

Jiayin Wang, Ph.D.

Academic Editor

PLOS ONE

Additional Editor Comments (optional):

I think all the issues suggested by reviewers have been solved.

Reviewers' comments:

Reviewer's Responses to Questions

**Comments to the Author**

1. If the authors have adequately addressed your comments raised in a previous round of review and you feel that this manuscript is now acceptable for publication, you may indicate that here to bypass the “Comments to the Author” section, enter your conflict of interest statement in the “Confidential to Editor” section, and submit your "Accept" recommendation.

Reviewer #3: (No Response)

Reviewer #4: All comments have been addressed

2. Is the manuscript technically sound, and do the data support the conclusions?

Reviewer #3: Partly

Reviewer #4: (No Response)

3. Has the statistical analysis been performed appropriately and rigorously? 

Reviewer #3: I Don't Know

Reviewer #4: (No Response)

4. Have the authors made all data underlying the findings in their manuscript fully available?

Reviewer #3: Yes

Reviewer #4: (No Response)

5. Is the manuscript presented in an intelligible fashion and written in standard English?

Reviewer #3: Yes

Reviewer #4: (No Response)

6. Review Comments to the Author

Reviewer #3: Thanks for your revision. I believe this manuscript can be published after a little modification.

1. Although you have deleted a large number of materials that may be unnecessary or redundant, this article is still too long to read. There are a lot of repetitive and subjective words, and some concluding remarks can be as concise as possible without too much explanation or description.

2. The structure of this paper should be reorganized. An article usually has no more than five parts, such as introduction, model, discussion and case study. I think it is necessary to combine some paragraphs into one part. It is strongly recommended to refer to other published manuscripts.

Reviewer #4: (No Response)

7. PLOS authors have the option to publish the peer review history of their article (what does this mean?). If published, this will include your full peer review and any attached files.

Reviewer #3: No

Reviewer #4: **Yes: **Yixuan Wang

---

## [Editor Report · Acceptance letter]

19 Nov 2021

PONE-D-20-24616R2 

Time-to-event estimation of birth prevalence trends:
a method to enable investigating the etiology of childhood disorders including autism 

Dear Dr. MacInnis:

I'm pleased to inform you that your manuscript has been deemed suitable for publication in PLOS ONE. Congratulations! Your manuscript is now with our production department. 

Kind regards, 

on behalf of

Dr. Jiayin Wang 

Academic Editor

PLOS ONE